# *OpenFly*: A Comprehensive Platform for Aerial Vision-Language Navigation

**Yunpeng Gao[1,2*], Chenhui Li[1*], Zhongrui You[1,3*], Junli Liu[1,2*], Zhen Li[1,4*],
Pengan Chen[1,5], Qizhi Chen[1,6], Zhonghan Tang[1,7], Liansheng Wang[1], Penghui Yang[1,8],
Yiwen Tang[1,2], Yuhang Tang[1,2], Shuai Liang[1,9], Songyi Zhu[1], Ziqin Xiong[1,9], Yifei Su[1,10],
Xinyi Ye[1], Jianan Li[1], Yan Ding[1], Dong Wang[1], Xuelong Li[11], Zhigang Wang[1†], Bin Zhao[1,2†]**

[1]Shanghai AI Laboratory, [2]Northwestern Polytechnical University,
[3]Beihang University, [4]Shanghai Jiao Tong University,
[5]The University of Hong Kong, [6]Zhejiang University,
[7]University of Science and Technology of China,
[8]East China University of Science and Technology, [9]Fudan University,
[10]Institute of Automation, Chinese Academy of Sciences, [11]TeleAI

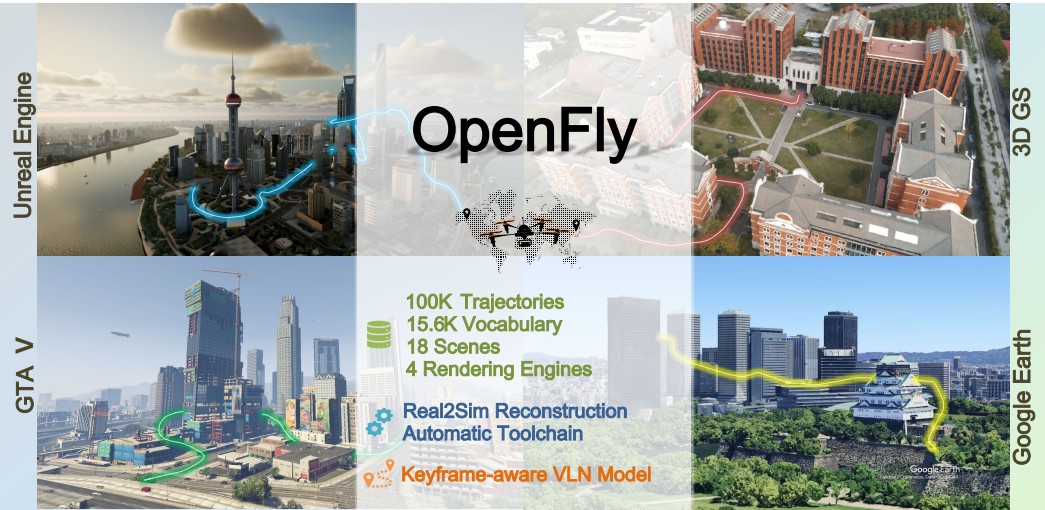

Figure 1: Overview of OpenFly. This work consists of (1) the integration of 4 rendering engines, significantly enhancing the diversity of scenario resources for aerial vision-language navigation; (2) an automatic data generation toolchain, eliminating reliance on labor-intensive annotations; (3) the largest aerial VLN dataset to date, comprising 100K trajectories; and (4) a keyframe-aware VLN model, achieving superior performance in both simulated and real-world scenes.

## Abstract

Aerial Vision-Language Navigation (VLN) seeks to guide UAVs by leveraging language instructions and visual cues, establishing a new paradigm for human-UAV interaction. However, the collection of VLN data demands extensive human effort to construct trajectories and corresponding instructions, hindering the development of large-scale datasets and capable models. To address this problem, we propose **OpenFly**, a comprehensive platform for aerial VLN. Firstly, OpenFly integrates 4 rendering engines and advanced techniques for diverse environment simulation, including Unreal Engine, GTA V, Google Earth, and 3D Gaussian Splatting (3D GS). Particularly, 3D GS supports real-to-sim rendering, further enhancing the realism of our environments. Secondly, we develop a highly automated toolchain for aerial VLN data collection, streamlining point cloud ac-

---

*Equal Contribution.
†Corresponding Author.

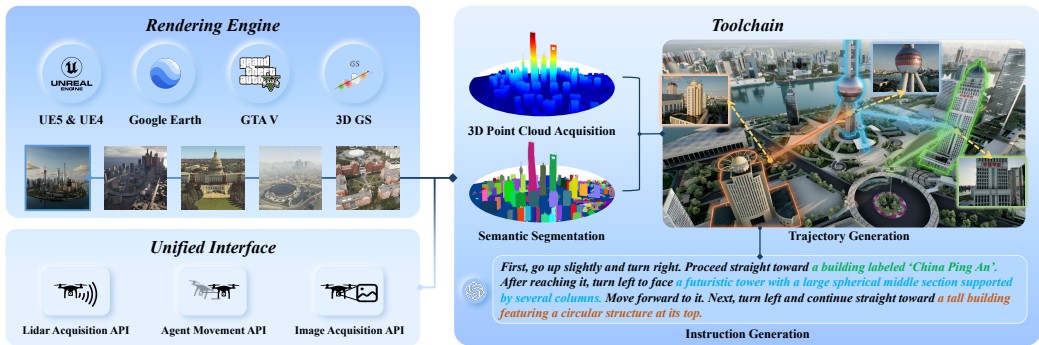

Figure 2: Framework of the automatic data generation. Multiple rendering engines are integrated to provide diverse, high-quality scenes. Built on these, several interfaces and tools are developed to enable automated generation of trajectories and instructions.

quisition, scene semantic segmentation, flight trajectory creation, and instruction generation. Thirdly, based on the toolchain, we construct a large-scale aerial VLN dataset with 100k trajectories, covering samples of diverse scenarios and assets across 18 scenes. Moreover, we propose OpenFly-Agent, a keyframe-aware VLN model emphasizing key observations to promote performance and reduce computations. For benchmarking, extensive experiments and analyses are conducted, where our navigation success rate outperforms others by 14.0% and 7.9% on the seen and unseen scenarios, respectively. The toolchain, dataset, and codes will be open-sourced.

# 1 INTRODUCTION

Embodied AI has drawn growing research attention, where vision-language navigation (VLN) emerging as a core task that navigate agents to a target location according to linguistic instructions and visual observations. A number of benchmark datasets have been established, *e.g.,* Touch-Down (Chen et al., 2019), REVERIE (Qi et al., 2020), R2R (Anderson et al., 2018), RxR (Ku et al., 2020), CVDN (Thomason et al., 2019), VLN-CE (Krantz et al., 2020), and LANI (Misra et al., 2018), which have significantly advanced the development of VLN methods (Long et al., 2024; Hong et al., 2022; Wang et al., 2024c; Chen et al., 2022; Zhang et al., 2024; Cao et al., 2025; Guo et al., 2025; Ma et al., 2025). Nevertheless, existing efforts primarily target indoor or ground-based agents, while unmanned aerial vehicles (UAVs), crucial for aerial photography, rescue operations, and cargo transport, remain unexplored.

Most recently, AerialVLN (Liu et al., 2023) and OpenUAV (Wang et al., 2024a) have made significant strides by leveraging UAV simulators to mitigate the scarcity of aerial VLN datasets, thereby driving advances in this field. However, several critical challenges remain to be addressed:

- **Limited data diversity.** Existing methods rely on AirSim and Unreal Engine (UE) for UAV control, which confines them to digital assets compatible with these platforms, limiting the diversity of available data and constraining the incorporation of more photorealistic sources.

- **High collection cost.** The process of generating trajectories relies on pilots operating UAVs in simulators, followed by manual annotation to create language instructions. The entire process is labor-intensive, time-consuming, and difficult to scale.

- **Small data scale.** Current datasets for aerial VLN remain relatively small, containing only about 10k trajectories, which is far behind embodied manipulation datasets. By contrast, Open X-Embodiment (O'Neill et al., 2024) and EO-1 (Qu et al., 2025) have collected over 1M episodes of manipulation, significantly promoting the development of vision-language-action (VLA) models.

To address these issues, we propose **OpenFly**, a comprehensive platform consisting of diverse rendering engines, a versatile toolchain, and a large-scale benchmark for the aerial VLN task. **To**

**enhance data diversity**, the platform is established on various widely-used rendering engines and advanced techniques, *i.e.,* UE, GTA V, Google Earth, and 3D Gaussian Splatting (3D GS), enabling us to utilize a wide range of assets as shown in Fig. 1. In particular, we use UAVs to capture numerous real-world images and integrate 3D GS technology into our platform to reconstruct realistic 3D scenes, empowering real-to-sim simulation. **To improve the efficiency of data collection**, we develop a versatile toolchain for automated aerial VLN data generation as depicted in Fig. 2. Specifically, point cloud acquisition is first conducted to capture the 3D occupancy of a scene. Next, scene semantic segmentation is performed to identify and select landmarks as waypoints along the flight trajectories. Building on these tools, trajectory generation is then carried out, taking landmarks and point clouds as input, using predefined flight actions as basic units, and automatically searching for a collision-free trajectory. Finally, we feed the trajectories and corresponding UAV-egocentric images into a vision-language-model (VLM), *e.g.,* GPT-4o, to generate linguistic instructions. The entire pipeline is highly automated, reducing the reliance on UAV pilots and annotators. **To collect a large-scale dataset**, we meticulously collected 18 high-quality scenes, generating various trajectories of differing heights and lengths. Benefitting from our toolchain, we are able to quickly construct a dataset of **100k** samples, significantly larger than existing datasets.

Besides, we propose **OpenFly-Agent**, a keyframe-aware aerial VLN model incorporating an adaptive frame-level sampling mechanism to emphasize critical observations containing instruction-related landmarks, leading to performance improvement and computation reduction compared to a uniform sampling strategy. Extensive experiments are conducted on the OpenFly dataset to evaluate numerous methods, establishing a comprehensive benchmark for the aerial VLN tasks. Overall, our contributions can be summarized as follows:

- We build OpenFly on multiple rendering engines and develop a versatile toolchain, enabling the automatic generation of data with high diversity and efficiency.

- We have constructed a large-scale aerial VLN benchmark comprising 100k trajectories across 18 high-quality scenes. To the best of our knowledge, this is the largest aerial VLN benchmark to date, and users can collect more customized data using the OpenFly platform.

- We propose OpenFly-Agent, a keyframe-aware VLN model. Extensive experiments in both simulated and real-world settings demonstrate its superior performance.

## 2 RELATED WORKS

### 2.1 VISION-LANGUAGE NAVIGATION DATASETS

Numerous datasets have been constructed to accelerate the VLN task. R2R (Anderson et al., 2018) focuses on evaluating agents in unseen buildings and provides discrete navigation options. RxR (Ku et al., 2020) provides a more densely annotated VLN dataset. TouchDown (Chen et al., 2019) and REVERIE (Qi et al., 2020) have each contributed a dataset from real-life environments, which requires a ground-based agent to follow instructions and find a target object. CVDN (Thomason et al., 2019) presents a cooperative VLN dataset where agents can access the history of human cooperation for inference. All the above datasets are graph-based, where navigable points are predefined. LANI (Misra et al., 2018) and VLN-CE (Krantz et al., 2020) propose the VLN task in continuous outdoor/indoor environments, enabling agents to move freely to any unobstructed point. Recently, a few works have tried to construct VLN datasets for aerial space. ANDH (Fan et al., 2022) establishes a dialogue-based aerial VLN dataset with bird-view images. CityNav (Lee et al., 2024) builds on the point cloud data from SensatUrban (Hu et al., 2022) and linguistic annotations from CityRefer (Miyanishi et al., 2023), which requires a real-world 2D map to help locate specific landmarks in the instruction. AerialVLN (Liu et al., 2023), OpenUAV (Wang et al., 2024a) and CityNav-Agent (Zhang et al., 2025) integrate AirSim and UE to create VLN scenes where pilots can control UAVs to generate various trajectories.

## 2.2 Vision-Language Navigation Methods

VLN methods enable agents to follow language instructions based on visual observations. Early approaches, such as graph-based methods (Ma et al., 2019; Wang et al., 2019; Ke et al., 2019; Fu et al., 2020), model the environment as a set of predefined nodes, with agents navigating between these discrete states. However, these methods are limited in dynamic, real-world environments. In recent years, LLM-driven approaches (Zhou et al., 2024b;a; Chen et al., 2024; Zeng et al., 2025) have utilized large language models to enhance reasoning and infer navigation steps, offering more flexibility in continuous environments. Despite significant progress, LLM-based methods still face challenges in grounding language instructions with real-world sensory data and adapting to unknown environments. Meanwhile, training-free LLM-based methods (Hu et al., 2025; Xu et al., 2025) provide a flexible way to infer navigation steps from language alone, enabling rapid adaptation of agents without retraining. In contrast, works like (Irshad et al., 2021; Krantz et al., 2021; Zhang et al., 2024; Song et al., 2025) have shifted focus to continuous spaces, aiming for more realistic navigation in dynamic settings. More recently, aerial VLN has gained attention, with AerialVLN (Liu et al., 2023) proposing a lookahead guidance method for better training trajectories, while STMR (Gao et al., 2024) enhances spatial reasoning through matrix representations, and OpenUAV (Wang et al., 2024a) integrates human feedback with ground-truth trajectories to guide navigation.

## 3 Automatic Data Generation

In this section, we first introduce the rendering engines and data resources, then present the developed toolchain. The overall framework for automatic data generation is illustrated in Fig. 2.

### 3.1 Rendering Engines and Data Resources

We leverage multiple rendering engines to construct diverse and realistic environments. Specifically, **Unreal Engine** provides eight urban scenes spanning over $100km^2$ with rich assets such as buildings, vehicles, and pedestrians. **GTA V** contributes a highly realistic cityscape modeled after Los Angeles. **Google Earth** offers four urban regions (Berkeley, Osaka, Washington D.C., and St. Louis) covering $53.60km^2$. Besides, hierarchical **3D Gaussian Splatting** (Kerbl et al., 2024) is employed for the reconstruction of real-world environments from UAV data, encompassing more than $7km^2$ across five campuses with diverse landmarks. More details and examples are provided in Appendix **??**.

### 3.2 Toolchain for Automatic Data Collection

To achieve automatic data generation, we first integrate the above rendering engines and design three unified interfaces to control the agent movement and acquire sensor data (presented in Appendix **??**). Based on these interfaces, we further develop a toolchain, streamlining point cloud acquisition, scene semantic segmentation, trajectory creation, and instruction generation.

**3D Point Cloud Acquisition.** OpenFly integrates various rendering engines and scenes, exhibiting distinct characteristics. To address these differences, we provide two methods to reconstruct the point cloud map for different scenes. 1) Rasterized Sampling Reconstruction: For UE and GTA V scenes, we customize rasterized sampling points at appropriate resolutions, followed by using the developed interface to obtain the local point cloud at the sampling points and stitch them for the entire scene. 2) Image-Based Sparse Reconstruction: In 3D GS, the scene reconstruction process begins with the open-source COLMAP (Schönberger & Frahm, 2016) framework, which generates a sparse point cloud from input images. We directly export and use the point clouds from this step.

**Scene Semantic Segmentation.** VLN requires meaningful landmarks as navigation targets. Thus, we offer three semantic segmentation methods to identify landmarks. 1) 3D Scene Understanding: A sequence of top-down views of the scene is captured in a rasterized format, followed by the off-the-shelf Octree-Graph (Wang et al., 2024b) to extract semantic 3D instances. 2) Point Cloud Projection and Contour Extraction: We acquire the point cloud of a scene and project the voxelized point cloud onto the ground. For each instance, its contour is segmented, and the maximum height of its points is used as the final height. Additionally, semantic annotations are obtained by feeding the segmented instances to GPT-4o for caption. 3) Manual Annotation: When the point cloud quality of a scene

Table 1: Comparisons of different VLN datasets. $N_{traj}$: the number of total trajectories. $N_{vocab}$: vocabulary size. Path Len: the average length of trajectories, measured in meters. Intr Len: the average length of instructions. $N_{act}$: the average number of actions per trajectory.

| Dataset | $N_{traj}$ | $N_{vocab}$ | Path Len. | Intr Len. | Action Space | $N_{act}$ | Environment |
|---|---|---|---|---|---|---|---|
| R2R (Anderson et al., 2018) | 7189 | 3.1K | 10.0 | 29 | graph-based | 5 | Matterport3D |
| RxR (Ku et al., 2020) | 13992 | 7.0K | 14.9 | 129 | graph-based | 8 | Matterport3D |
| TouchDown (Chen et al., 2019) | 9326 | 5.0K | 313.9 | 90 | graph-based | 35 | Google Street View |
| VLN-CE (Krantz et al., 2020) | 4475 | 4.3K | 11.1 | 19 | 2 DoF | 56 | Matterport3D |
| AerialVLN (Liu et al., 2023) | 8446 | 4.5K | 661.8 | 83 | 4 DoF | 204 | AirSim + UE |
| CityNav (Lee et al., 2024) | 32637 | 6.6K | 545 | 26 | 4 DoF | - | SensatUrban |
| OpenUAV (Wang et al., 2024a) | 12149 | 10.8K | 255 | 104 | 6 DoF | 264 | AirSim + UE |
| Ours | 100K | 15.6K | 99.1 | 59 | 4 DoF | 35 | AirSim + UE, GTA V, Google Earth Studio, 3D GS + SIBR viewers |

is low or finer segmentation is required, OpenFly provides an interface for manually annotating instances and semantics within the point cloud. Users can choose these methods flexibly based on their requirements. The corresponding details and results are shown in Appendix **??**.

**Automatic Trajectory Generation.** Leveraging the point cloud map and segmentation tools, Open-Fly can automatically generate VLN trajectories using the following method. First, a global voxel map $M_{global}$ is constructed from the scene point cloud. Second, a landmark is randomly chosen as the target, with a starting point being selected at a certain distance from the landmark, and a point close to the landmark being chosen as the endpoint. Third, A collision-free trajectory is generated using the A* (Hart et al., 1968) pathfinding algorithm based on $M_{global}$ and a customized action space. By repeatedly selecting the endpoint as the new starting point, complex trajectories can be generated. Finally, utilizing OpenFly's interface, UAV-egocentric images corresponding to the trajectory points are obtained as visual observations. More details are included in Appendix **??**.

**Automatic Instruction Generation.** Most previous works have predominantly relied on manual annotation to generate instructions, which is costly and hinders dataset scalability (Liu et al., 2023; Lee et al., 2024; Liu et al., 2025). To address this issue, we propose a highly automated instruction generation method based on VLMs, *e.g.,* GPT-4o.

A straightforward method would be to input all images to VLMs to analyze the trajectory and generate instructions. However, using all images introduces considerable computational overhead and causes significant difficulties for a VLM to understand. Additionally, we find the 'Forward' action usually occupies a larger proportion of a flight trajectory, with 'Turn Left/Turn Right' or 'Ascend/Descend' actions taken when encountering key landmarks. Therefore, we split the complete trajectory into multiple sub-trajectories according to action transitions, extracting key actions and images for processing. Notably, slight angle adjustments often occur during flight to change direction subtly, which will be ignored in this procedure. We submit the action sequence and the last captured three images of each sub-trajectory to a VLM to generate a sub-instruction of both the action and the landmark. All sub-instructions of the same trajectory are then processed by an LLM to integrate into a complete instruction. The proposed strategy significantly improves the instruction accuracy compared to directly inputting all trajectory images to a VLM. To further verify the data quality, we randomly select 3K samples from the entire dataset according to the data distribution in Sec. 4.2. After manually inspecting these samples, we find that they reach a high qualification rate of 91%. The problematic data involves some vague descriptions, but it is still considered acceptable by examiners. Besides, all test data have undergone manual inspection, with low-quality ones removed. Thanks to GPT's high concurrency, we can quickly generate a large number of instructions, which solves the problem of difficult and time-consuming manual annotation. **More details of instruction generation and data quality control are provided in Appendix ?? and ??.**

## 4 DATASET ANALYSIS

Table 1 summarizes key statistics of several commonly used VLN datasets, from which we can see that our dataset features a significantly larger number of trajectories and a greater environmental diversity. In contrast, our average trajectory length and instruction length are relatively short. This

is intentional, we argue that short- and medium-range instructions better reflect natural human usage habits and may be more beneficial for advancing aerial VLN.

## 4.1 TRAJECTORY AND INSTRUCTION ANALYSIS

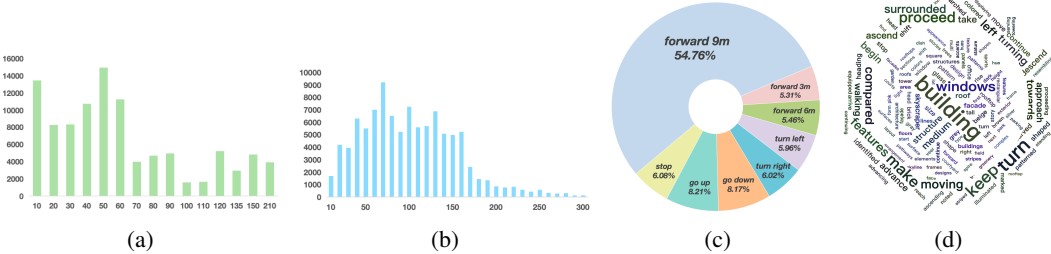

Figure 3: Statistical analysis of the generated data. (a) Height distributions of trajectories. (b) Length distributions of trajectories. (c) Action distributions. (d) Word cloud of verbs and nouns.

Using our toolchain, we collect a dataset of 100K trajectories, which is much larger than other aerial VLN datasets. Compared with ground-based VLN, the aerial VLN task has more motion dimensions. Therefore, we set different trajectory lengths and flight heights to obtain rich data. Fig. 3a and 3b exhibit the distribution of these data, with their lengths ranging from 0 to 300 meters, and the heights ranging from 0 to 210 meters. Notably, we follow the mainstream methods (Krantz et al., 2020; Liu et al., 2023) to use discrete actions, *e.g.,* 'Forward' and 'Turn left', for trajectory generation, where the step size of the 'Forward' action is set to 3 m, 6 m, and 9 m to adapt to targets at different distances. Fig. 3c presents the action distribution of our dataset. It should be noted that collected trajectories also provide corresponding waypoint information, which can be further processed into smoother trajectories to enable navigation waypoint prediction. Besides, the OpenFly platform supports trajectory generation with continuous waypoints directly based on drone trajectory planning algorithms Wang et al. (2022); Zhou et al. (2021); Mellinger & Kumar (2011); Zhou et al. (2019). To further enhance data diversity, we incorporate the DAgger Ross et al. (2011) algorithm as a data augmentation functionality. In summary, OpenFly offers a comprehensive platform that allows users to generate custom data on their own. It also supports agent interaction and enables real-time retrieval of both agent poses and environmental data. This makes it compatible with On-policy training approaches.

For instruction analysis, the vocabulary size of our dataset is 15.6K, and the average length of instructions is 59. Fig. 3d illustrates the word clouds of nouns and verbs, where 'building', 'windows', and 'skyscraper' are the most common references, and 'proceed' and 'turn' are the mostly used verbs for VLN. Due to the space limitation, **we put more details in Appendix ??.**

## 4.2 DATASET SPLIT

Similar to previous works, we divide the dataset into three splits, *i.e., Train, Test Seen, Test Unseen*. For the *Train* split, 7 scenes under the UE rendering engine account for 75.7% of all data, since UE provides the largest number of scenes, where different amounts of trajectories are sampled according to the areas of scenes. The 4 scenes created by 3D GS are also the main part of the data, accounting for nearly 20% of the total amount. To ensure visual quality, we only collect data from a high-altitude perspective using Google Earth, which accounts for 4.46%. The *Test Seen* data consists of 1800 trajectories uniformly sampled from 11 seen scenarios, and the *Test Unseen* data comprises 1200 trajectories uniformly generated from 3 unseen scenes, *i.e.,* UE-smallcity, 3D GS-sjtu02, and a Los Angeles-like city in GTA V. Detailed data distributions are shown in Appendix **??.**

## 5 OPENFLY-AGENT

Fig. 4 illustrates the architecture of our OpenFly-Agent, an aerial VLN model that builds upon the OpenVLA (Kim et al., 2024) baseline, since OpenVLA and aerial VLN share a similar pipeline, *i.e.,* taking images and instructions as input and generating actions. OpenVLA is trained on 1M

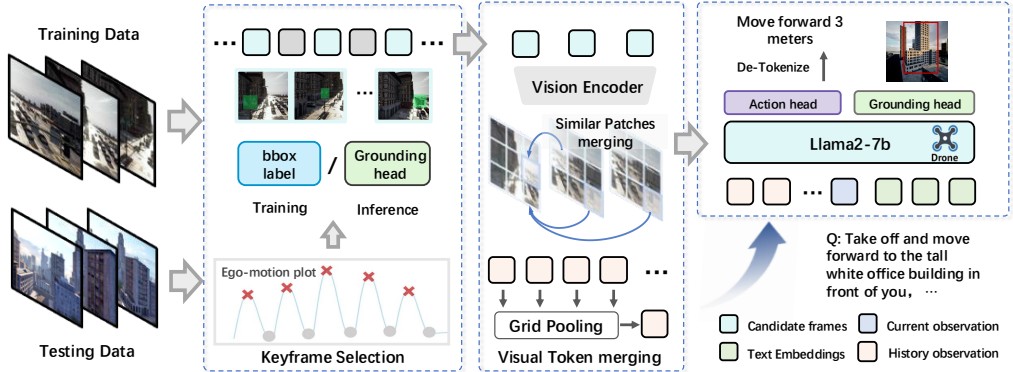

Figure 4: The architecture of OpenFly-Agent. Keyframes are selected according to action transitions and the landmark grounding module to extract crucial observations as the history, with corresponding visual tokens compressed to further reduce the computational burden.

data, having strong abilities in instruction-following and reasoning, which establishes an efficient initialization for our model. In contrast, our OpenFly-Agent takes a sequence of images as input to indicate the observation history instead of one image in the original OpenVLA. Additionally, to mitigate visual redundancy between adjacent video frames while maintaining key information, two strategies are designed, *i.e.,* keyframe selection and visual token merging. First, a series of candidate keyframes is selected based on the UAV flight trend and a landmark grounding module. Then, these keyframes are merged temporally, resulting in a compact sequence of visual tokens. Finally, the action decoder discretizes the predicted tokens to 6 action types specific to UAVs.

## 5.1 KEYFRAME SELECTION

The length of contextual visual tokens is a major challenge for VLMs when processing videos. Many open-source VLMs use uniform frame sampling (Buch et al., 2022; Ranasinghe et al., 2024; Wang et al., 2025) to reduce calculation, but this strategy is not suitable for aerial VLN, since it may miss frames containing key landmarks. To address this issue, a keyframe selection strategy is proposed to emphasize important visual observations. We notice that sudden changes in the UAV's trajectory are often caused by the observation of landmarks, which can serve as a kind of cues to determine keyframes. Therefore, a heuristic method is adopted to select candidate frames by identifying the change point of the UAV's movement, followed by extracting the corresponding frame and two frames before and after it from the trajectory, constituting a keyframe set. Moreover, we design a landmark grounding module, which consists of three cross-attention layers to incorporate text and image features from the LLM hidden state, predicting the bounding boxes $b \in \mathcal{R}^4$ of the instruction-indicated landmark. To incorporate as many landmark-related regions as possible into the historical visual tokens, candidate frames with the bounding boxes' area greater than the threshold $\theta$ will be retained as the final keyframes. During the training process, we obtain the bounding box of each landmark using the developed tools introduced in Sec. 3.2, enabling the training of the grounding module and the accurate selection of keyframes. During the testing process, the bounding box of each frame is sequentially estimated by the well-trained grounding module. Then, our model selects keyframes by bounding boxes area and adjacent frames when a significant motion change occurs, forming a keyframe set for this moment.

## 5.2 VISUAL TOKEN PRUNING

To further reduce redundant information in keyframes, we introduce visual token merging into OpenFly-Agent. For the keyframes selected by the above method, a visual encoder maps them to multiple visual tokens, with each token representing the information of an image patch. Considering the potential inter-frame patch redundancy, we take a strategy that similar tokens in adjacent frames are periodically merged. Specifically, we select the frame with the largest bounding box in a keyframe set as the reference, since it usually contains the crucial observation indicating the landmark in an instruction. Then, we densely calculate the cosine similarities between each pair

of visual tokens of the reference image and other comparative images in a keyframe set. Next, we merge the tokens with high similarity by averaging them, with the unmerged tokens in the comparative frame being discarded. The merging operation is iteratively performed until the entire keyframe set has been traversed. Besides, we maintain a memory bank with a capacity of $K$ images, following a first-in-first-out (FIFO) policy to retain the latest keyframes. Since aerial VLN requires UAVs to perform long-distance flights based on instructions, we continue to conduct token compression within each keyframe to reduce the computational burden. The compressed visual tokens are obtained through grid pooling (Li et al., 2024). Notably, we keep the visual tokens of the current frame uncompressed to capture the latest visual observation, as it contains the most important information for action prediction.

## 6 EXPERIMENTS

### 6.1 IMPLEMENTATION AND TRAINING DETAILS

The proposed OpenFly-Agent adopts the OpenVLA (Kim et al., 2024) as the baseline, with the current frame during flight remaining 256 tokens and all historical keyframes compressed into 1 token. The capacity $K$ of the history memory bank is set to 2 in our experiment. For the action head, the last 256 tokens in the vocabulary are used as special tokens for action representation. Similar to (Liu et al., 2023; Lee et al., 2024), 6 actions for UAVs are defined as {Forward, Turn Left, Turn Right, Move Up, Move Down, Stop}. The OpenFly-Agent is trained with a batch size of 64 and a learning rate of 2e-5. The grounding module is optimized with a GIoU loss function, and the threshold $\theta$ for keyframe selection is set to 0.25 times the size of the input image.

### 6.2 EVALUATION METRICS

Four standard metrics in VLN tasks are adopted to evaluate different methods, *i.e.,* navigation error (NE), success rate (SR), oracle success rate (OSR), and success weighted by path length (SPL). NE measures the average deviation between the UAV's final stopping point and the ground-truth destination. SR calculates the proportion of successful tasks, where a task is considered successful if the UAV stops within 20 m of the target (Liu et al., 2023). Each environment provides corresponding point clouds that enable collision checking. If a collision occurs, the task is counted as a failure. In OSR, if any point on the trajectory is within 20 m of the target, the task can be considered successful. SPL calculates the success rate weighted by the ratio of the ground-truth path length to the actually-executed path length.

### 6.3 QUANTITATIVE RESULTS

We evaluate the proposed OpenFly-Agent and multiple VLN methods on the test set, with quantitative results listed in Table 2, where Seq2Seq, CMA, and AerialVLN achieve limited success rates. In contrast, Navid (Zhang et al., 2024) and NaVila (Cheng et al., 2024) are two most recent VLN methods, obtaining better results and demonstrating the great potential of VLMs in aerial VLN. See-Point-Fly Hu et al. (2025) is a zero-shot method, which is evaluated using GPT-4.1 as the agent and demonstrates reasonable robustness. Our OpenFly-Agent outperforms the comparison methods by a large margin, benefiting from the proposed strategies. While aerial VLN is an emerging and challenging task, and there is still much room for improvement. The results on the test-unseen split indicate the generalization abilities of these methods. Similarly, our method achieves the best performance, exhibiting a certain degree of robustness. However, all methods are significantly degraded, indicating that more powerful models are urgently needed to be developed.

### 6.4 REAL-WORLD EXPERIMENTS

The real-world experiments are conducted in 23 real outdoor scenes, where each scene corresponds to an unseen VLN task created by human operators, and the trajectory lengths range from 50m to 500m. We use a Q250 airframe as a real agent, carrying an NVIDIA Jetson Xavier NX running Ubuntu 18.04 as the onboard computer. In the real-world experiments with the drone, we utilize the "Super" (Ren et al., 2025) trajectory planning framework for local trajectory planning and employ Model Predictive Control (MPC) Falanga et al. (2018) for trajectory tracking. The advantage of

Table 2: Comparison results on the test set. 'Random' means randomly selecting one action to execute until the 'stop' action is chosen. All models are retrained using our dataset.

| Method | test-seen | | | | test-unseen | | | |
|---|---|---|---|---|---|---|---|---|
| | NE↓ | SR↑ | OSR↑ | SPL↑ | NE↓ | SR↑ | OSR↑ | SPL↑ |
| Random | 242m | 0.7% | 0.8% | 0% | 301m | 0.1% | 0.1% | 0% |
| Seq2Seq (Krantz et al., 2020) | 205m | 2.9% | 24.3% | 2.6% | 229m | 2.1% | 20.6% | 1.1% |
| CMA (Krantz et al., 2020) | 161m | 5.4% | 28.1% | 4.8% | 217m | 4.6% | 24.4% | 2.1% |
| See-Point-Fly (Hu et al., 2025) | - | - | - | - | 191m | 8.2% | 12.7% | 6.3% |
| AerialVLN (Liu et al., 2023) | 139m | 7.5% | 30.0% | 6.8% | 214m | 7.3% | 28.1% | 4.4% |
| Navid (Zhang et al., 2024) | 153m | 13.0% | 38.2% | 11.6% | 210m | 10.8% | 27.2% | 5.0% |
| NaVila (Cheng et al., 2024) | 132m | 20.3% | 53.5% | 17.8% | 202m | 14.7% | 42.1% | 9.6% |
| OpenFly-Agent (Ours) | **93m** | **34.3%** | **64.3%** | **24.9%** | **154m** | **22.6%** | **56.2%** | **19.1%** |

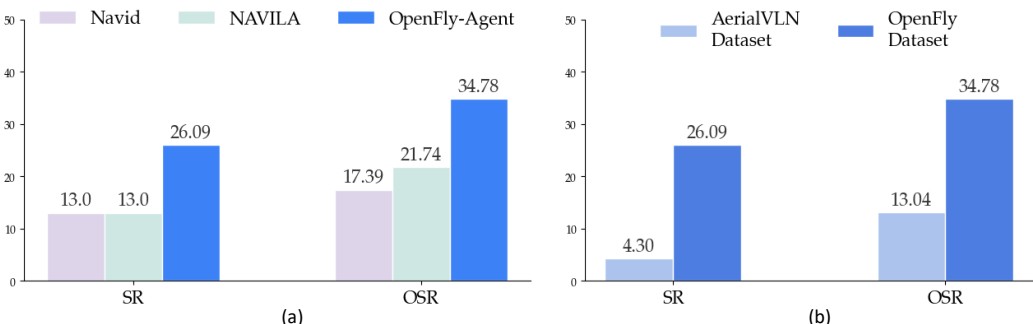

Figure 5: Results of real-world experiments. (a) Comparison with two strong VLN methods. (b) Performances of OpenFly-Agent trained on different datasets.

this paradigm is that it enables the VLN model to adapt to various planning and control algorithms, thereby accommodating diverse robotic platforms and scenarios. All methods run on an external PC communicating with the onboard computer to transfer images and action instructions. Two most recent models, Navid (Zhang et al., 2024) and NaVila (Cheng et al., 2024), are evaluated for comparison. The results are shown in Fig 5 (a), where our model achieves the best performance with 26.09% SR and 34.78% OSR, significantly outperforming the comparison methods. This experiment again indicates the superiority of our OpenFly-Agent. Besides, we also trained our model on both our own dataset and the AerialVLN dataset separately. The results are shown in Fig. 5 (b), strongly demonstrating the capability of our data generation method in bridging the sim-to-real gap. A qualitative result is presented in Fig. 6, and a dynamic demo can be found in our supplementary video.

## 6.5 ABLATION STUDY

Ablation studies are conducted to evaluate the contribution of the keyframe selection and visual token merging in OpenFly-Agent. Table. 3 shows the results, where OpenVLA (Kim et al., 2024) is our baseline. Using only the current frame or uniformly selecting from previous observation as keyframes makes the model perform poorly in the aerial VLN task. From 'History + VTM', we can see that historical frames significantly improve the success rate. The keyframe selection strategy further increases the SR from 16.6% to 34.3%, demonstrating the effectiveness of key observations. Besides, the comparison between 'KS' and 'KS + VTM' indicates the great effect of our visual token merging strategy. We find that there is a severe imbalance between the number of text and image tokens if the token merging strategy is not applied. The cross-modal signal can be diluted by the numerical imbalance between a few text tokens and many visual tokens Luo et al. (2025). As a result, background clutter, task-irrelevant distractors, and environmental noise may be encoded indiscriminately, leading to excessive computational cost and diluted attention to task critical cues Li et al. (2026).

Instruction: Fly straight towards that campus-looking building with the red roof. Once closed, give a little left turn until you spot the river right in front of you. Then, keep flying forward until you see the whole bridge on the river.

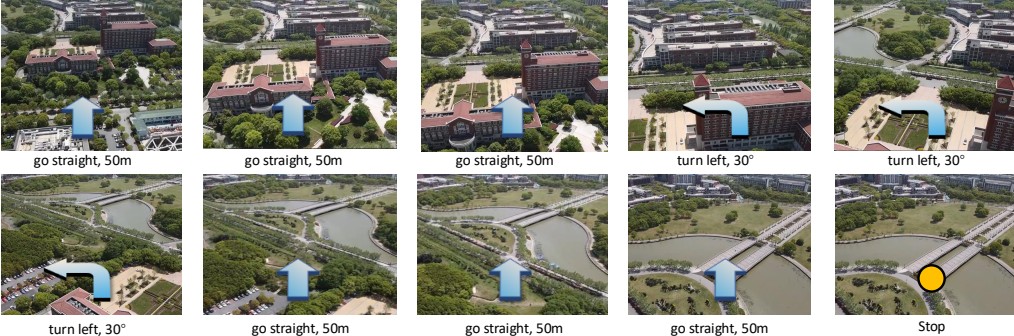

Figure 6: Snapshots of the real-world experiment.

## 7 CONCLUSION

In this work, we present OpenFly, a platform designed for large-scale data collection in aerial Vision-and-Language Navigation (VLN). Open-Fly integrates multiple rendering engines and provides an automatic toolchain for data generation, enabling efficient collection of diverse, high-quality aerial VLN data. The resulting large-scale dataset comprises 100k trajectories across 18 distinct scenes, spanning a wide range of altitudes and lengths, which is significantly larger than existing ones. Furthermore, we propose OpenFly-Agent, a keyframe-aware aerial VLN model capable of identifying frames with critical observations, leading to accurate flight action prediction.

Table 3: Ablation study on the test-seen split. 'KS' and 'VTM' denote keyframe selection and visual token merging, respectively. 'History' indicates uniform frame sampling. 'Random KS' means randomly selecting a frame from the candidate keyframe set.

| Method | NE↓ | SR↑ | OSR↑ | SPL↑ |
|---|---|---|---|---|
| OpenVLA (baseline) | 231m | 2.3% | 10.8% | 2.2% |
| History | 223m | 6.9% | 23.3% | 5.6% |
| Random KS | 264m | 8.7% | 26.6% | 5.8% |
| KS | 275m | 9.2% | 28.1% | 6.1% |
| History + VTM | 215m | 16.6% | 40.5% | 9.1% |
| KS + VTM | 93m | 34.3% | 64.3% | 24.9% |

Extensive experiments validate the effectiveness of the proposed method, and establish a comprehensive benchmark for future advancements in aerial VLN.

## 8 ACKNOWLEDGEMENT

This work is supported by Shanghai AI Laboratory.

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
