# OpenReview forum: "OpenFly: A COMPREHENSIVE PLATFORM FOR AERIAL VISION-LANGUAGE NAVIGATION"
_ICLR.cc/2026/Conference — ICLR 2026 Poster_

### Official Review · Reviewer_6BXg · 2025-10-15

**Soundness:** 2
**Presentation:** 3
**Contribution:** 3
**Rating:** 6
**Confidence:** 5

**Summary:**

The authors introduce a dataset and method for aerial Vision–Language Navigation (VLN). The OpenFly dataset contains 100K automatically generated trajectories, synthesized using multiple rendering sources such as Unreal Engine and Google Earth. Compared to existing aerial VLN datasets, OpenFly is the largest in terms of trajectory count. The proposed OpenFly-Agent is a keyframe-aware model based on the OpenVLA framework. By selecting representative frames and merging redundant visual tokens from high-frequency inputs, OpenFly-Agent efficiently predicts navigation actions using a Llama2-7B backbone. Experimental results demonstrate significant improvements compared to existing VLN baselines, such as NaVila  (from 14.2% to 32.2% success rate on seen scenarios).

**Strengths:**

The OpenFly dataset's scale (100K trajectories) represents a significant contribution despite its automated generation. Additionally, the performance of OpenFly-Agent is noteworthy, as it substantially outperforms recent baselines.

**Weaknesses:**

A primary limitation of the paper is its limited technical novelty. Since the model builds upon the existing OpenVLA framework, the novelty mainly lies in incremental enhancements, specifically the addition of keyframe selection and token pruning mechanisms. Several experimental issues raise further concerns:

(1) How were the Navid and NaVila baselines trained?

Implementation details and training procedures for these baselines are not provided in the paper. This makes it challenging to validate whether OpenFly-Agent indeed outperforms them in a fair comparison.

(2) Why was OpenVLA selected as the backbone?

The performance of OpenVLA baseline (2.3% in Table 3) is significantly lower than that of Navid (13.0%) and NaVila (20.3%) in Table 1. Given this substantial gap, the authors' choice of OpenVLA as the backbone over more performant alternatives such as NaVila requires further justification.

(3) Does OpenFly-Agent generalize to other datasets?

Considering that trajectories are automatically generated, direct comparisons in terms of dataset scale to handcrafted datasets such as TouchDown and CityNav become less relevant. Rather, the automated data-generation process could be viewed as part of the training methodology itself. Thus, evaluations on external datasets (e.g., AerialVLN, CityNav, OpenUAV) are necessary to substantiate the generalization capability of OpenFly-Agent.

**Questions:**

Please refer to the questions listed under Weaknesses. I like the approach of this paper. However, clarification and a generalization evaluation are needed. The paper itself is well-written and easy to follow.

---

> ### Author Response · Authors · 2025-11-22
> **Response to Reviewer 6BXg**
>
> > **W1: How were the Navid and NaVila baselines trained?**
>
>  For fairness, all comparison models were fine-tuned on our dataset. Specifically, we replaced the actions in our dataset with text actions and corresponding parameters to fit the text output framework of Navid and Navila. In addition, all other hyperparameters followed the original settings of the open-source codes of comparison algorithms.
>
> > **W2: Why was OpenVLA selected as the backbone?**
>
> We aim to validate the effectiveness of our design with a concise baseline model. Since Navid and NaVila adopt more complex architectures, we instead choose the simpler OpenVLA to keep the evaluation focused and interpretable.
>
> > **W3: Does OpenFly-Agent generalize to other datasets? e.g., AerialVLN, CityNav, OpenUAV**
>
> Thanks for the suggestion. We have conducted extra validations on other feasible datasets.
> - AerialVLN: To evaluate the model's robustness on the AerialVLN dataset, we trained and evaluated our model using its data. The performance on the val_unseen dataset is summarized below:
>
>
> | Method    | NE↓  | SR↑    | OSR↑   | SPL↑  |
> |-----------|------|--------|--------|-------|
> | AerialVLN | 128m | 5.10%  | 10.50% | 1.40% |
> | Ours      | 96m  | 12.50% | 17.20% | 1.90% |
>
> It can be seen that our model significantly outperforms the AerialVLN model, demonstrating the generalizability of our OpenFly-Agent. However, the performance is lower than that in our OpenFly dataset. We analyzed the results and found that the AerialVLN dataset is very challenging and contains a considerable amount of noisy data, such as missing landmarks (e.g., “go straight, slow down slowly, and turn a little left”) and ambiguous landmarks (e.g., “go to the building”). The extremely high difficulty and the vague instructions result in a relatively low success rate for all methods.
>
>
> - CityNav: Direct comparison is not feasible due to fundamental incompatibilities:
>   1. Map Dependency: CityNav requires additional map priors, whereas OpenFly-Agent operates in a strictly map-free setting.
>   2. Altitude & Visual Domain Gap: CityNav is designed for high-altitude navigation using point-cloud-based simulation. This rendering mechanism produces low-resolution RGB images near buildings, creating a significant visual domain gap that fails to support the fine-grained perception required for our task.
>
> - OpenUAV: Comparison is unsuitable due to fundamental paradigm mismatches. OpenUAV requires ground-truth trajectory assistance during execution to handle occlusions, whereas OpenFly operates in a fully vision-only paradigm. This core difference makes direct cross-dataset evaluation impractical.
>
> We consider improving cross-dataset generalization across more heterogeneous platforms as an important direction for future work.

---

### Official Review · Reviewer_oJcE · 2025-10-25

**Soundness:** 3
**Presentation:** 3
**Contribution:** 3
**Rating:** 4
**Confidence:** 5

**Summary:**

This proposes a large-scale dataset for aerial visual-language navigation.  The dataset comprises a high diversity of simulation scenes collected from 4 rendering engines, and it annotates a large number of navigation trajectories using autmated toolchain.  This paper demonstrates that the proposes dataset enhances performant robot policies.  It trains an OpenVLA model on the collected trajectories, which shows stronger performance on in-domain scenes and better generalization to out-of-domain scenes, during testing.  The authors commit that the dataset and codes will be publicly released.

**Strengths:**

1. This paper is well written.  It illustrates clearly the limitations of previous methods and how it addresses them in the newly proposed dataset.
2. The proposed dataset contains a high diversity of scenes (18 in total), while some correspond to real-world scenes
3. This paper proposes a strong baseline based on the performance OpenVLA model, outperforming other baselines by a large margin
4. The trained OpenVLA agent shows strong generalization to unseen scenes during testing
5. The trajectory collection pipeline is neat, which generates navigation trajectories without the need of manual effort

**Weaknesses:**

1. Although this paper provides details evaluation of OpenVLA agents, it lacks analysis on the effectiveness of the proposed dataset. In other words, this paper does not address the question "Do the diversity and quality of collected navigation trajectories facilitate more performant navigation agents, compared to other datasets?". I'd recommend the authors train OpenVLA on both OpenFly and OpenUAV datasets, and test on unseen scenes to highlight the effectiveness of the proposed dataset.
2. This paper utilizes A* algorithm to generate navigation trajectory.  However, this paper does not comment or provide any analysis on the potential biased introduced by A* algorithm.  For example, are the generated trajectories smooth, energy efficient, or safe? These are key factors that should be considered when deploying simulation-trained agents to the real world.
3. Likewise, this paper selects 4 DoF action space, which sounds reasonable for long-horizon navigation.  However, such a simplified action space is insufficient for drones to navigate in highly-occluded scenes (e.g. street scenes).  This paper doesn't discuss such limitations.
4. I'm not sure if training OpenVLA models for navigation is a right reseaerch direction, since recent studies show that navigation tasks are inherently visual grounding tasks.  Since VLMs excel at visual grounding, one can directly use VLM models to generate 2D waypoints for 3D navigation [1].  I'd strongly recommend the authors to test these zero-shot navigation policies on the proposed dataset.

---

Reference:
[1] Hu, Chih Yao, et al. "See, Point, Fly: A Learning-Free VLM Framework for Universal Unmanned Aerial Navigation." Conference on Robot Learning. PMLR, 2025.

**Questions:**

1. Do the diversity and quality of collected navigation trajectories facilitate more performant navigation agents, compared to other datasets?  A comparison of OpenVLA models trained on OpenFly and OpenUAV and tested on unseen scenes is needed.
2. What are the biases introduced by A* algorithm for generating action trajectories?
3. What are the limitations of 4 DoF action space in aerial navigation tasks?
4. What is the performance of VLM-based navigation models (See-Point-Fly) on the proposed dataset?

I'm open to increase my rating if the authors are able to address these concerns.

---

> ### Author Response · Authors · 2025-11-22
> **Response to Reviewer oJcE**
>
> > **W1: Train OpenVLA on both OpenFly and OpenUAV datasets, and test on unseen scenes to highlight the effectiveness of the proposed dataset.**
>
> Thanks for the suggestion. First, fundamental paradigm mismatches make OpenUAV unsuitable for direct comparison.
> OpenUAV requires ground-truth trajectory assistance and specific numerical descriptions for target search. In contrast, OpenFly operates without any GT aid, relying solely on trajectory descriptions and visual features. This core difference makes direct cross-dataset evaluation impractical.
>
> Second, we verified the effectiveness of the proposed dataset by using the AerialVLN dataset as a fairer baseline. AerialVLN shares the same vision-only setting as OpenFly, making it the appropriate benchmark. We demonstrated OpenFly's superiority in Figure 5(b) of the original manuscript. Specifically, we trained the same model (OpenFly-Agent) separately on OpenFly and AerialVLN datasets, and evaluated them across 23 unseen real outdoor scenes. The results show that the model trained on OpenFly outperforms the AerialVLN-trained counterpart, demonstrating our dataset's quality and its capability in bridging the sim-to-real gap.
>
> > **W2: Are the generated trajectories smooth, energy efficient, or safe?**
>
> The generated trajectories are relatively smooth (with the max single turn angle to 30°), energy-efficient and safe (by the shortest collision-free path planning).
>
> > **W3:  A simplified action space is insufficient for drones to navigate in highly-occluded scenes (e.g. street scenes).**
>
> For aerial VLN tasks in large-scale urban environments, our discretized action space (move forward 3 m, turn ±30°, shift left/right 3 m, ascend 2 m, and descend 2 m) is sufficient to accomplish the mission. This set of actions adequately covers the motion primitives required for typical urban navigation tasks, such as progressing along streets or corridors, adjusting heading at intersections, performing lateral corrections, and handling moderate altitude changes.
>
> Moreover, we adopted a hierarchical scheme to achieve navigation in real-world scenes, where our OpenFly-Agent focuses on high-level understanding and prediction, and traditional UAV algorithms are used for low-level planning and control. This design can handle navigation in general outdoor scenarios. It is noteworthy that unlike RL-based navigation and avoidance methods designed for densely cluttered environments, our goal is vision-and-language navigation in large-scale outdoor scenes, where the emphasis is on understanding language instructions and the environment, and on predicting high-level paths.
>
> > **W4: I'd strongly recommend the authors to test these zero-shot navigation policies on the proposed dataset.**
>
> Thanks for the suggestion. Following your recommendation, we evaluated the zero-shot navigation strategy "See, Point, Fly" on our proposed OpenFly dataset. Under the same experimental setup as described in the original paper, we used GPT-4.1 as the agent and obtained a success rate of approximately 8% on OpenFly dataset. Considering that this result is achieved under a fully zero-shot setting and within significantly more complex environments and tasks, it already demonstrates reasonable effectiveness and a certain level of robustness on our benchmark. The results have been added in Table 2 in the revised paper.
>
> Further failure-case analysis shows that in some simulated environments, GPT-4.1 often fails to reliably recognize key landmarks in the scene, which leads the agent to "spin in place". Even when the model successfully identifies the target object, the generated action sequence can still be misaligned with the correct direction. These observations are consistent with the current community consensus that VLMs still have notable limitations in fine-grained spatial understanding and spatial reasoning. At the same time, they also validate the effectiveness of the OpenFly dataset in exposing and amplifying such bottlenecks, indicating that our benchmark can provide valuable evaluation grounds and research insights for future improvements in zero-shot navigation methods and VLMs.

---

> > ### Comment · Reviewer_oJcE · 2025-11-23
> >
> > Thank the authors for the response.  My major concerns have been addressed.

---

### Official Review · Reviewer_HbYF · 2025-10-29

**Soundness:** 3
**Presentation:** 3
**Contribution:** 2
**Rating:** 6
**Confidence:** 3

**Summary:**

The paper "OpenFly: A Comprehensive Platform for Aerial Vision-Language Navigation" presents OpenFly, a novel platform aimed at addressing the challenges in aerial vision-language navigation (VLN) for UAVs (unmanned aerial vehicles). It tackles the issues of limited data diversity, high data collection costs, and small dataset scales that have hindered previous aerial VLN efforts. The key contributions of this paper are:
1.Platform Design: OpenFly integrates four rendering engines (Unreal Engine, GTA V, Google Earth, and 3D Gaussian Splatting), significantly enhancing the diversity of simulated environments used for VLN tasks. The 3D Gaussian Splatting technique, in particular, allows for realistic real-to-sim rendering, making the environments more photorealistic.
2.Automated Data Generation: The paper introduces a toolchain for automating the collection of aerial VLN data, which includes steps like point cloud acquisition, scene semantic segmentation, trajectory creation, and instruction generation. This toolchain reduces the need for manual annotation, thus enabling the large-scale generation of high-quality data.
3.Large-Scale Dataset: OpenFly's platform results in a large-scale dataset consisting of 100,000 trajectories, making it one of the largest aerial VLN datasets to date. This dataset spans 18 different scenes and includes a wide variety of environmental features and flight scenarios.

**Strengths:**

1. Innovative Data Generation Platform:
   OpenFly integrates four rendering engines (Unreal Engine, GTA V, Google Earth, and 3D Gaussian Splatting), which enhances the diversity of training environments for aerial vision-language navigation (VLN). This combination provides a wide range of realistic simulation environments for training models.
2. Automated Data Generation Toolchain:
   The platform features an automated toolchain for data collection, semantic segmentation, trajectory generation, and instruction creation. This reduces the reliance on manual annotations and makes it easier to scale data collection processes, allowing the creation of large-scale VLN datasets.
3. Real-World Experimentation and Model Performance:
   OpenFly-Agent is tested in real-world scenarios, and the model shows strong performance in both simulated and unseen environments, highlighting its potential for real-world application in aerial navigation tasks.

**Weaknesses:**

1. Why use such an old model, Llama2-7b, as the baseline?
2. The paper cites OpenUAV, but why isn't there a comparison with their method?
3. VTM appears to be a pooling layer, but there's no explanation for why the performance improvement is so significant.
4. Are keyframes selected based on rules? What would be the difference if they were selected uniformly? Would the performance decrease if important frames are missed?

**Questions:**

same as weakness.

---

> ### Author Response · Authors · 2025-11-22
> **Response to Reviewer HbYF**
>
> > **W1: Why use such an old model, Llama2-7b, as the baseline?**
>
> First, Llama2-7b is a classic and concise architecture that we aim to use to demonstrate the effectiveness of the proposed method. Second, we adopted the structure of Llama2-7b but did not use its original parameters. The version trained on large-scale internet data is utilized to further train our OpenFly-Agent, allowing it to acquire a broader and more diverse understanding of language and visual concepts.
>
> > **W2: The paper cites OpenUAV, but why isn't there a comparison with their method?**
>
> Although OpenUAV addresses the same UAV vision-and-language navigation problem, its task formulation and experimental setup are not directly comparable to ours. OpenUAV needs auxiliary obstacle avoidance assistance or requires access to the ground-truth trajectory.
>
> For fair comparison, we adapted the OpenUAV model to our data and setup for reference. Specifically, we removed the Assistant module and also removed the prompt content describing specific numerical values (e.g., "The target is at a yaw angle of 45 degrees from you"). Under this adapted setting, we retrain the OpenUAV model on our data, achieving only a 2.01% success rate on our test set. Our analysis indicates that the model fails to recognize landmarks and eventually gets stuck rotating in place.
>
>
> > **W3: VTM appears to be a pooling layer, but there's no explanation for why the performance improvement is so significant.**
>
> The token merging strategy (VTM) can help maintain a relatively balanced number of text tokens and image tokens, which is crucial for our task. There will be a severe imbalance if the VTM is not applied. In this case, we observe a "less is more" phenomenon where excessive frames may degrade performance due to context dilution [3][4]. Text prompts convey high-level intent but under-specify pixel-aligned details, the cross-modal signal can be diluted by the numerical imbalance between a few text tokens and many visual tokens[5]. As a result, background clutter, task-irrelevant distractors, and environmental noise may be encoded indiscriminately, leading to excessive computational cost and diluted attention to task critical cues [6].
>
> To verify the above points, we conducted extra visualization experiments on the role of VTM, demonstrating the model can focus on the most relevant information with VTM. The results have been added to the Figure 13 of the revised paper.
>
>
> > **W4-1: Are keyframes selected based on rules?**
>
> Keyframes are selected based on a heuristic rule and learned modules. Specifically, we design and train a Landmark Grounding module to identify frames that contain landmarks, ensuring that important visual cues are recognized and preserved. After keyframe selection, the VTM module further compresses and refines the information.
>
> > **W4-2:What would be the difference if they were selected uniformly? Would the performance decrease if important frames are missed?**
>
> Thanks for the suggestion. The new experiments are shown in the Table below, where "History" indicates uniformly sampling image as keyframe, and "Random KS" means randomly selecting a frame from the candidate keyframe set. They all have a higher probability of missing key information, and degrade the success rate (SR).
>
> **Table: Ablation study on the test-seen split. `KS` and `VTM` denote keyframe selection and visual token merging, respectively.**
>
> | Method               | NE↓  | SR↑   | OSR↑  | SPL↑  |
> |----------------------|------|-------|-------|-------|
> | OpenVLA (baseline)   | 231m | 2.3%  | 10.8% | 2.2%  |
> | History              | 223m | 6.9%  | 23.3% | 5.6%  |
> | Random KS            | 264m  | 8.7%  | 26.6% | 5.8%  |
> | KS                   | 275m | 9.2%  | 28.1% | 6.1%  |
> | History + VTM        | 215m | 16.6% | 40.5% | 9.1%  |
> | KS + VTM             | 93m  | 34.3% | 64.3% | 24.9% |
>
>
> **Reference**
>
> [1]Ren Y, Zhu F, Lu G, Cai Y, Yin L, Kong F, Lin J, Chen N, Zhang F. Safety-assured high-speed navigation for MAVs. Science Robotics. 29;10(98):eado6187, 2024.
>
> [2]Falanga D, Foehn P, Lu P, Scaramuzza D. PAMPC: Perception-aware model predictive control for quadrotors. In IEEE/RSJ International Conference on Intelligent Robots and Systems (IROS), pp. 1-8, 2018.
>
> [3]Wang, S., Chen, Z., Xu, Y., Guo, W., & Xiong, H. Less is More: Token-Efficient Video-QA via Adaptive Frame-Pruning and Semantic Graph Integration. arXiv preprint arXiv:2508.03337.
>
> [4]Li, Y., Wang, C., & Jia, J. Llama-vid: An image is worth 2 tokens in large language models. In European Conference on Computer Vision, pp. 323-340, 2024.
>
> [5]Luo, J., Lin, J., Zhang, Z., Wu, B., Fang, M., Chen, L., & Tang, H. Univid: The open-source unified video model. arXiv preprint arXiv:2509.24200.
>
> [6]Li, W., Zhang, R., Shao, R., Fang, Z., Zhou, K., Tian, Z., & Nie, L. SemanticVLA: Semantic-Aligned Sparsification and Enhancement for Efficient Robotic Manipulation. (Accepted to AAAI 2026)

---

### Official Review · Reviewer_6nDA · 2025-10-29

**Soundness:** 3
**Presentation:** 3
**Contribution:** 2
**Rating:** 4
**Confidence:** 4

**Summary:**

This paper presents OpenFly, an Aerial VLN platform addressing data scarcity. It integrates four rendering engines (using 3D GS for Real-to-Sim) and an automated data toolchain to generate a 100k-trajectory dataset. A novel keyframe-aware model, OpenFly-Agent, achieves SOTA performance in simulation and 23 real-world scenarios, validating its strong sim-to-real transfer.

**Strengths:**

1.	The paper directly tackles the most significant bottleneck in Aerial VLN. The automated toolchain is a highly valuable and practical contribution, drastically lowering the barrier to data collection.
2.	The 100k-trajectory dataset is the largest to date. More importantly, the integration of four distinct rendering engines, especially the use of 3D GS for real-world reconstruction, ensures exceptional environmental diversity and realism.
3.	The OpenFly-Agent, with its keyframe-aware selection and token merging, presents an intelligent approach to handling redundant video data in VLN, improving both efficiency and performance.

**Weaknesses:**

1. Relying on the A* data-generation pipeline introduces a two problems: (1) The path style is unnatural, filled with "robotic sharp turns" instead of smooth, human-like flight. (2) The data is pure "expert demonstration", meaning the model only learns to follow perfect paths, not recover from deviations. This makes the model extremely fragile to real-world disturbances.
2. The system uses discrete motion control, which in practice, is more like performing control classification based on visual input. During actual deployment, the UAV is unable to achieve precise flight maneuvers, such as orbiting a small object, because the current turning angles are insufficient

**Questions:**

1. The paper mentions in Section 3.2 that collision-free training trajectories are generated using A*. However, during the simulation and real-world testing in Section 6, no collision detection mechanism or related metrics (e.g., number of collisions) are mentioned. Please clarify: (1) During evaluation, are the agent's generated trajectories checked for collisions? (2) If a collision occurs, is the task considered a failure, or does it continue? (3) Does the OpenFly-Agent itself possess any form of dynamic or reactive collision avoidance capability?

2. Could the authors clarify the exact movement mechanism for the agent in the simulation? The description suggests the model chooses between a series of discrete waypoints or actions (e.g., Forward 9m). This raises a critical question: after executing an action, does the agent "teleport" to the next state, or does it execute a simulated continuous flight action between points? If it is "teleportation," the task essentially degenerates from a "navigation control" problem into a "sequential VQA" problem. This is fundamentally different from the real world, where a UAV must use continuous motor control to counteract inertia, wind, and various physical disturbances. Please clarify this setup, as it profoundly impacts the sim-to-real difficulty and the model's true generalization capabilities.

3. The ablation study shows a massive performance leap when moving from "KS" (9.2% SR) to "KS + VTM" (34.3% SR), suggesting the VTM module is critical. Could the authors elaborate on the reasons why this token merging strategy results in such a dramatic improvement?

4. How to deal with the problem in weakness one?

I think the paper's overall contribution to be valuable, particularly the development of a pipeline for building UAV environments. Even if the trajectories have certain issues, the continuous videos are still useful for non-control applications, such as VLM pretraining or developing world models.
However, the paper is positioned as a comprehensive 'platform.' This implies that in addition to the environment, the trajectories and control mechanisms must also be sound and well-justified.
Therefore, I expect the authors to specifically address my questions on these points. I will consider raising my score upon receiving a satisfactory response.

---

> ### Author Response · Authors · 2025-11-22
> **Response to Reviewer 6nDA Part 1**
>
> >**W1-1: Relying on the A\*  data generation pipeline will cause the path style is unnatural, filled with "robotic sharp turns".**
>
>  We set the hyperparameter for the single turn angle to 30° to avoid sharp turns during the path generation process. This is consistent with the action space of our model.
>
>
>
> >**W1-2:  The data is pure "expert demonstration", which may make the model fragile to real-world disturbances.**
>
> This is a common issue of imitation learning, but we adopted a series of measures to generate diverse data, enhancing the robustness and generalization capability of the model.
>
> For example, our data covers different heights ranging from 10 meters to 210 meters, different trajectory lengths ranging from 10 meters to 300 meters, and 18 large-scale distinct scenarios. As shown in Figure 5(b) of our original manuscript, we compare the real-world performance of OpenFly-Agent trained on the AerialVLN dataset versus our dataset (4.30 SR vs. 26.09 SR), demonstrating that our dataset provides a substantial improvement in generalization.
>
>
>
>
> > **W2: The UAV is unable to achieve precise flight maneuvers, such as orbiting a small object.**
>
> For VLN tasks involving UAVs operating in large-scale urban environments, a discretized action space (move forward 3 m, turn ±30°, shift left/right 3 m, ascend 2 m, and descend 2 m) is sufficient to accomplish the mission. This set of actions adequately covers the motion primitives required for typical urban navigation tasks, such as progressing along streets or corridors, adjusting heading at intersections, performing lateral corrections, and handling moderate altitude changes. Moreover, we manually sampled and inspected the generated trajectories, and found them to be reasonable and consistent with expected UAV navigation behaviors.
>
> Regarding "orbiting a small object", this behavior can in fact be achieved through a sequence of actions
> such as move forward, turn left (or right), move forward, turn left (or right), and so on. However, such
> tasks are relatively uncommon in typical UAV navigation applications.
>
> Besides, we delegate the low-level flight maneuvers to traditional UAV planning and control algorithms (using the "Super" [1] for local trajectory planning and Model Predictive Control (MPC) [2] for trajectory tracking). The advantage of this paradigm is that it enables the VLN model to adapt to various low-level planning and control algorithms, thereby accommodating diverse robotic platforms and scenarios. We have clarified this point in the "REAL-WORLD EXPERIMENTS" section of the revised manuscript.

---

> ### Author Response · Authors · 2025-11-22
> **Response to Reviewer 6nDA Part 2**
>
> > **Q1-1: During evaluation, are the agent's generated trajectories checked for collisions?  If a collision occurs, is the task considered a failure, or does it continue?**
> >
> Each environment provides corresponding point clouds that enable collision checking. If a collision occurs, the task is counted as a failure. We have updated and clarified the description in the "EVALUATION METRICS" section of the revised manuscript.
>
>
> > **Q1-2: Does the OpenFly-Agent itself possess any form of dynamic or reactive collision avoidance capability?**
>
> The OpenFly-Agent has the ability to avoid large obstacles (e.g., buildings) through its high-level policy. For small obstacles, collision avoidance is handled by the low-level planning and control (as mentioned in the above W2, where we describe the integrated design of high-level action prediction and low-level control).
>
> > **Q2: After executing an action, does the agent "teleport" to the next state, or does it execute a simulated continuous flight action between points?**
>
> Actions are executed as continuous flight using an off-the-shelf low-level control algorithm.
>
> > **Q3: Why this token merging strategy results in such a dramatic improvement?**
>
> The token merging strategy (VTM) can help maintain a relatively balanced number of text tokens and image tokens, which is crucial for our task. There will be a severe imbalance if the VTM is not applied. In this case, we observe a "less is more" phenomenon where excessive frames may degrade performance due to context dilution [3][4]. Text prompts convey high-level intent but under-specify pixel-aligned details, the cross-modal signal can be diluted by the numerical imbalance between a few text tokens and many visual tokens[5]. As a result, background clutter, task-irrelevant distractors, and environmental noise may be encoded indiscriminately, leading to excessive computational cost and diluted attention to task critical cues [6].
>
> To verify the above points, we conducted extra visualization experiments on the role of VTM, demonstrating the model can focus on the most relevant information with VTM. The results have been added to the Figure 13 of the revised paper.
>
>
> **Reference**
> [1]Ren Y, Zhu F, Lu G, Cai Y, Yin L, Kong F, Lin J, Chen N, Zhang F. Safety-assured high-speed navigation for MAVs. Science Robotics. 29;10(98):eado6187, 2024.
>
> [2]Falanga D, Foehn P, Lu P, Scaramuzza D. PAMPC: Perception-aware model predictive control for quadrotors. In IEEE/RSJ International Conference on Intelligent Robots and Systems (IROS), pp. 1-8, 2018.
>
> [3]Wang, S., Chen, Z., Xu, Y., Guo, W., & Xiong, H. Less is More: Token-Efficient Video-QA via Adaptive Frame-Pruning and Semantic Graph Integration. arXiv preprint arXiv:2508.03337.
>
> [4]Li, Y., Wang, C., & Jia, J. Llama-vid: An image is worth 2 tokens in large language models. In European Conference on Computer Vision, pp. 323-340, 2024.
>
> [5]Luo, J., Lin, J., Zhang, Z., Wu, B., Fang, M., Chen, L., & Tang, H. Univid: The open-source unified video model. arXiv preprint arXiv:2509.24200.
>
> [6]Li, W., Zhang, R., Shao, R., Fang, Z., Zhou, K., Tian, Z., & Nie, L. SemanticVLA: Semantic-Aligned Sparsification and Enhancement for Efficient Robotic Manipulation. (Accepted to AAAI 2026)

---

> > ### Comment · Reviewer_6nDA · 2025-11-26
> >
> > 1. Thank you for the authors' reply. First, regarding the expert trajectory, I still haven't seen the response I was looking for. Simply increasing the trajectory altitude or the complexity of the environment/scenario, while still generating trajectories using the A* algorithm, does not fundamentally change the approach. This has no relation to Imitation Learning (IL). The imitation data can also be perturbed. What I was actually hoping to see was a response that addressed trajectory robustness using methods like On-Policy training or DAgger, which is crucial for real-world deployment. I urge the authors to seriously consider that even a slight gust of wind can significantly impact the result when deployed on a real system. I believe the overall scope of work in your paper is acceptable, but these implementation details are lacking.
> >
> > 2. My continuous emphasis on the ability to rotate around small obstacles stems from the belief that the current discrete control strategy is a suboptimal solution. Similar to autonomous driving, the most ideal state is to use depth information to predict a waypoint (or sequence of waypoints). The movement between these waypoints can then be executed using your current control model. This approach would inherently offer a much higher degree of freedom in the completed maneuvers. To put it another way: what is the latency required for the drone to switch from flying forward to flying backward? It might require $360^{\circ} / 30^{\circ} = 12$ steps, whereas a waypoint-based approach would only require a single step. Given such a good simulation environment, it would be an enormous improvement if you could incorporate continuous control.

---

> > > ### Author Response · Authors · 2025-11-27
> > > **Response to Reviewer 6nDA**
> > >
> > > We greatly appreciate your constructive feedback, which highlights key considerations for real-world deployment.
> > >
> > > (1)	We would like to clarify that although our model does not utilize On-policy training, the OpenFly platform's simulation environment supports agent interaction and enables real-time retrieval of both agent poses and environmental data. This makes it compatible with On-policy training approaches.
> > >
> > > (2)	Additionally, the OpenFly platform provides a comprehensive set of tools that make it straightforward to implement the DAgger method. For example, during data collection, the agent executes actions according to its current policy, and at each step, the agent’s actions are recorded. Simultaneously, expert actions are generated using path-planning algorithm based on the "current state to the target." If the discrepancy between the agent’s action and the expert’s action exceeds a threshold, the expert’s action is adopted until the target is successfully reached. The combined dataset can then be used for further training.
> > >
> > > Following your suggestion, **we have quickly implemented and supported the DAgger algorithm, and this functionality will also be open-sourced.** In the revised manuscript, we have expanded the "Automatic Trajectory Generation" section to provide a more detailed explanation of the DAgger method and On-policy training support.

---

> > > ### Author Response · Authors · 2025-11-27
> > > **Response to Reviewer 6nDA Part2**
> > >
> > > Thanks a lot for your suggestion regarding continuous waypoints.
> > >
> > > (1)	We would like to clarify that, as mentioned in the "TRAJECTORY AND INSTRUCTION ANALYSIS" section of the original manuscript, **"OpenFly platform also supports trajectory generation with continuous waypoints"** directly based on drone trajectory planning algorithms [1-4]. We offer a comprehensive platform that allows users to generate custom data on their own.
> > >
> > > (2)	To align with main-stream methods (e.g., AerialVLN，Navid), we adopted a discrete action modeling approach. However, the collected trajectories not only include discrete actions but also corresponding waypoint information, which can be further processed into smoother and more coherent trajectories. We have provided this functionality in our platform, and made a clarification in the "TRAJECTORY AND INSTRUCTION ANALYSIS" section of the revised manuscript.
> > >
> > >
> > > Thanks again for your suggestions, which are valuable in helping us clarify details and improve the quality of the manuscript. **All tools, data, and models on our platform will be open-sourced**, and we believe that OpenFly will contribute to the development of the Aerial VLN field. We would greatly appreciate it if you could support our work and improve the score.
> > >
> > > **References**：
> > >
> > > [1] Z. Wang, X. Zhou, C. Xu and F. Gao, "Geometrically Constrained Trajectory Optimization for Multicopters," in IEEE Transactions on Robotics, vol. 38, no. 5, pp. 3259-3278, Oct. 2022, doi: 10.1109/TRO.2022.3160022
> > >
> > > [2] Zhou, Xin, et al. "Ego-planner: An esdf-free gradient-based local planner for quadrotors." IEEE Robotics and Automation Letters 6.2 (2020): 478-485.
> > >
> > > [3] Mellinger, Daniel, and Vijay Kumar. "Minimum snap trajectory generation and control for quadrotors." 2011 IEEE international conference on robotics and automation. IEEE, 2011.
> > >
> > > [4] Zhou, Boyu, et al. "Robust and efficient quadrotor trajectory generation for fast autonomous flight." IEEE Robotics and Automation Letters 4.4 (2019): 3529-3536.

---

### Author Response · Authors · 2025-11-22
**Rebuttal Summary**

**Dear Area Chairs and Senior Area Chairs,**
we sincerely appreciate your time and effort in reviewing our manuscript. The following is a brief summary of our paper and the rebuttal.

We propose OpenFly, a comprehensive platform for aerial VLN, integrating 4 rendering engines for diverse environment simulation and developing an automated toolchain for data collection. Based on the toolchain, we construct the largest aerial VLN dataset to date with **100k** trajectories. Moreover, we propose OpenFly-Agent, a keyframe-aware VLN model, outperforming SOTA works by **14.0%** and **7.9%** success rates on the seen and unseen scenarios, respectively.

___
**Strengths Recognized by Reviewers:**
1. The OpenFly automated toolchain is a highly valuable and practical contribution, significantly lowering the cost and barrier of large-scale data collection. (**6nDA,HbYF,oJcE,6BXg**)
2. The OpenFly dataset offers large-scale, high-quality, diverse data enabled by multiple rendering/reconstruction engines, including 3D Gaussian Splatting. (**6nDA,HbYF,oJcE,6BXg**)
3. OpenFly-Agent serves as a strong baseline, where keyframe-aware selection and visual token merging effectively improve efficiency and performance. The real-world experiments highlight its strong generalization ability. (**6nDA,HbYF,oJcE,6BXg**)
___
**Major Concerns and Brief Responses:**
1. Q: How do you address **trajectory robustness**? (**6nDA**)
    A: Our data covers different heights, trajectory lengths, and 18 large-scale distinct scenarios to enhance the data diversity. Besides, the OpenFly-Platform supports both **DAgger and on-policy training** paradigms mentioned by reviewer **6nDA**.
2. Q: Why do you choose **discretized actions** ? (**6nDA, oJcE**)
    A: For aerial VLN tasks in large-scale urban environments, we explained that **our discretized action space is sufficient to accomplish navigation goals**. Moreover, the **collected trajectories include not only discrete actions but also associated waypoint information**, which can be further processed into smoother and more coherent trajectories.
3. Q: What makes the **VTM module** effective, and how are **keyframes** selected? (**6nDA, HbYF**)
    A: We explained and conducted experiments to verify the importance of the VTM module and the keyframe selection strategy. (More details can be found in the response to reviewer **6nDA**, Q3.)
4. Q: How do you **validate the effectiveness** of the OpenFly dataset and the **generalization ability** of      OpenFly-Agent? (**oJcE, 6BXg**)
    A: We trained the OpenFly model separately on the OpenFly dataset and the AerialVLN dataset. Experiments show that the **OpenFly dataset leads to +21.79% success rate (Figure 5b of the original manuscript), validating its effectiveness**. We also trained OpenFly-Agent on the AerialVLN dataset, outperforming the AerialVLN model by **7.4%** success rate on the unseen test set.
5. Q: Comparison with **zero-shot navigation methods**. (**oJcE**)
    A: We compared our model with the zero-shot navigation method (see-point-fly) mentioned by reviewer **oJcE**. Our OpenFly-Agent outperforms see-point-fly by **14.6%** success rate on the unseen test set.
6. Q: Why choose **OpenVLA** as the baseline? (**6BXg**)
    A: We chose OpenVLA as a concise and clean baseline to clearly validate the effectiveness of our design.


___
**Summury:**
We engaged in active discussions with reviewers. More questions and details can be found in the following responses.

**Reviewer 6nDA (Initial Rating 4)**: This reviewer stated that **"I think the paper's overall contribution to be valuable, I will consider raising my score upon receiving a satisfactory response."** Due to policy constraints, the reviewer cannot provide further comments on the latest discussion.

**Reviewer HbYF (Initial Rating 6)**: No response yet.

**Reviewer oJcE (Initial Rating 4)**: The reviewer stated that **"I'm open to increase my rating if the authors are able to address these concerns."** After discussion, the reviewer claimed **"My major concerns have been addressed."**

**Reviewer 6BXg (Initial Rating 6)**: No response yet.

---

**All of OpenFly's tools, data, and models will be open-sourced to promote the development of the aerial VLN field.**

---

### Meta-Review · Area_Chair_ev4j · 2026-01-06

**Summary:**

This paper presents OpenFly, a comprehensive platform for aerial Vision-Language Navigation (VLN) that features an automated toolchain designed to significantly lower the barrier to large-scale data collection. The authors introduce a massive dataset comprising 100k trajectories integrated with advanced rendering engines like 3D Gaussian Splatting (3DGS). Furthermore, the proposed OpenFly-Agent incorporates keyframe-aware selection and Visual Token Merging (VTM), achieving a substantial performance leap over current state-of-the-art (SOTA) methods. While reviewers initially recognized the practical value of the platform, they raised concerns regarding the robustness of trajectories, the choice of a discretized action space, and the specific technical justification for the VTM module.

**Reviewer Concerns:**

The rebuttal phase successfully addressed the primary technical concerns raised by Reviewers 6nDA, oJcE, HbYF, and 6BXg. Regarding the discretized action space and trajectory robustness, the authors clarified its suitability for large-scale urban navigation and demonstrated the platform's capability to generate smooth, waypoint-based paths. Concerns regarding the dataset’s effectiveness and the agent's generalization were mitigated through additional comparative experiments, which showed a +21.79% success rate improvement when using the OpenFly dataset. Moreover, the authors provided technical evidence for the VTM module and offered a clear rationale for the chosen baselines, including comparisons with zero-shot navigation methods.

**Reviewer Scores:**

Following the discussion phase, Reviewer oJcE acknowledged the resolution of all primary concerns and expressed a willingness to upgrade the assigned rating. Regarding Reviewer 6nDA, the authors provided appropriate and comprehensive clarifications during the rebuttal that effectively addressed the technical doubts raised in the initial review. Meanwhile, Reviewers HbYF and 6BXg maintained positive assessments, reinforcing the overall consensus on the work's significance. Given that the authors have demonstrated the framework's technical soundness and committed to open-sourcing the tools, data, and models, the AC believes this work will be a highly valuable asset to the aerial robotics and embodied AI communities. Therefore, the AC recommends the Acceptance of the manuscript.

---

### Decision · Program_Chairs · 2026-01-26

Accept (Poster)